# Implementing active surveillance for TB—The views of managers in a resource limited setting, South Africa

Febisola I. Ajudua[1,2]*, Robert J. Mash[1]

1 Department of Family and Emergency Medicine, Division of Family Medicine and Primary Care, Faculty of Medicine and Health Sciences, Stellenbosch University, Cape Town, Western Cape, South Africa,
2 Department of Family Medicine and Rural Health, Faculty of Health Sciences, Walter Sisulu University, Mthatha, Eastern Cape, South Africa

* ajuduaf@sun.ac.za, febijudus@gmail.com

## Abstract

### Background

The achievement of the World Health Organization's END TB goals will depend on the successful implementation of strategies for early diagnosis and retention of patients on effective therapy until cure. An estimated 150,000 cases are missed annually in South Africa. It is necessary to look at means for identifying these missed cases. This requires the implementation of active surveillance for TB, a policy adopted by the National Department of Health.

### Aim

To explore the views of managers of the TB program on the implementation of active surveillance for TB in the resource constrained setting of the Eastern Cape, South Africa.

### Methods

A descriptive, explorative, thematically analysed qualitative study based on 10 semi-structured interviews of managers of the TB program. Interviews were transcribed verbatim and analysed using the framework method and Atlas-ti.

### Results

Active case finding of people attending health facilities was the dominant approach, although screening by community health workers (CHWs) was available. Both government and non-government organisations employed CHWs to screen door to door and sometimes as part of campaigns or community events. Some CHWs focused only on contact tracing or people that were non-adherent to TB treatment. Challenges for CHWs included poor coordination and duplication of services, failure to investigate those identified in the community, lack of transport and supportive supervision as well as security issues. Successes included expanding coverage by government CHW teams, innovations to improve screening, strategies to improve CHW capability and attention to social determinants.

**Data Availability Statement:** Transcripts of the interviews cannot be made publicly available. The interviewees have not given permission for the transcripts to be made publicly available. The reasons are 1. There were only ten interviewees. 2.

The content of the transcripts easily identify the interviewees and this contradicts the commitment made to maintain confidentiality of the interviewees. The transcripts will be made available to named researchers who require these transcripts for further research. These can correspond directly with the corresponding author with their requests for access to the transcripts. Email - febijudus@gmail.com The institutional point of contact email address for data availability is the Vice Dean: Research at Stellenbosch University, Cape Town, South Africa - researchfhs@sun.ac.za.

**Funding:** The funding for this study is from the South African National Research Foundation's Collaborative Postgraduate Training Grant received in 2017 by the Departments of Family Medicine at Stellenbosch University, Walter Sisulu University and the University of Limpopo. The funder played no role in the study design, data collection and analysis or the decision to publish.

**Competing interests:** The authors have declared that no competing interests exist.

## Conclusion

A multifaceted facility- and community-based approach was seen as ideal for active surveillance. More resources should be targeted at strengthening teams of CHWs, for whom this would be part of a comprehensive and integrated service in a community-orientated primary care framework, and community engagement to strengthen community level interventions.

## Introduction

The World Health Organization (WHO) currently estimates that 3.5 out of 10 active tuberculosis (TB) cases are missed globally [2]. In sub-Saharan Africa, this estimate rises to 5 out of 10 active TB cases [1]. South Africa is numbered among the countries with the highest burden of TB and TB is the leading cause of death [2–4]. The achievement of the END TB goals will depend on the successful implementation of strategies for active surveillance as well as retention of patients on treatment [5].

The National TB Program (NTP) in South Africa continues to develop policy to address the persistent problem of "the missing TB patients" [6,7]. The epidemic is prevalent in poorer, vulnerable and overcrowded communities and health seeking behavior varies considerably between individuals [8,9]. There is a need for community-based initiatives geared towards early identification and linkage to care for people with active TB [10]. Such initiatives should reduce the incidence of TB, rather than just treating those that come to facilities for treatment, while missing the hidden epidemic in communities [11].

Patients' pathway analysis identified a number of gaps occurring early on in the patient care cascade; [12] such as the inadequacy of strategies to identify presumptive cases of TB in communities, early loss to follow up before initiation of therapy and high rates of non-adherence to therapy. In the past decade, the NTP has implemented mainly facility-based strategies for early diagnosis of active TB and improved treatment regimens [13]. Although laudable in themselves, these initiatives require supporting strategies for active identification of suspected TB cases in the community [14,15].

The National Department of Health (NDoH) has adopted active surveillance as a policy in high risk populations, [13,16] in order to reduce transmission in communities [17,18]. Passive case-finding is not enough to address the challenge of reducing transmission in high burden settings such as South Africa [19–21]. Research from sub-Saharan countries affirms that active surveillance contributes to early diagnosis in high TB burden settings and invariably reduces transmission in the community [22,23].

This NDoH policy on TB is embedded within the broader re-engineering of primary health care in South Africa that is expected to improve health outcomes [16,24]. Health reform relies heavily on a community-oriented primary care (COPC) approach and the provision of community health workers (CHW). COPC shifts the focus of primary care services from the practice population to the population at risk [25–27]. COPC can include health promotion, disease prevention and screening, to enable early diagnosis of infectious diseases such as HIV and TB [16]. COPC may also include the use of CHWs to address social as well as health problems in these communities [28]. Therefore, the implementation of a COPC approach can provide an opportunity for implementation of active TB surveillance.

The WHO guidelines on systematic screening for TB advise that each country design programs for active surveillance based on its context [29]. The implementation of active surveillance for TB requires that due consideration be given to the prevailing 'influencing factors' in

the community [30]. There is still a paucity of literature on successful implementation of active surveillance for TB, although cost has been noted as a major issue in several countries [31].

The Eastern Cape Province is historically one of the most disadvantaged provinces in South Africa and continues to battle with underdevelopment and resource constraints. For an improved understanding of the challenges to implementation of active surveillance for TB in this resource constrained context, it was necessary to explore the views of managers of the TB program in the Province. The opinion of managers about active surveillance and the priority it should be afforded in ending the TB epidemic will help understand the contextual enablers and challenges unique to this context.

## Method

### Study design

This was a descriptive, exploratory, thematically analysed, qualitative study using semi structured interviews to explore the perspectives of TB managers. The Health Research Ethics Committee of Stellenbosch University approved this study, reference number S17/10/189_1243.

### Setting

The Eastern Cape in South Africa is a largely rural province divided into six districts and two metropolitan areas (Nelson Mandela Metropole and the Buffalo City Metropole). It is regarded as historically disadvantaged because much of the province was designated a homeland under Apartheid and was underdeveloped. The population in the Province is the third largest in the country, speaks mostly isiXhosa, and is estimated at 6.8 million people [32]. The median annual household income is $850, which is about half the South African average [33].

The Eastern Cape has the highest incidence of TB, which was reported as 692 per 100,000 in 2015 [34]. Screening for TB symptoms in facilities is below the national average (65% for $\geq$ 5 years of age with national average of 73%, 50% for < 5 years of age with national average of 65%). Initiation on treatment is the lowest in the country at 82% (national average 91%). Co-infection of TB with HIV is higher than the national average at 97% (national average 89%). TB treatment success rate is similar to the national average at 83% (national average 82%) and loss to follow up is on par with the national average at 7% [35].

The TB program in the Province is managed by a hierarchy of 12 Provincial and District TB managers. The TB Control Program is under a Provincial Program Manager who supervises three managers that are responsible for the portfolios of Drug Sensitive TB, Drug Resistant TB and Advocacy, Communication and Social Mobilisation (ACSM). In each District and Metropole, there is a manager responsible for managing the TB program. At the time of this study, there was a transition from TB managers (responsible only for TB) to HAST (HIV/AIDS, Sexually Transmitted Infections and TB) managers. In a number of these Districts the person remained the same even though the portfolio of responsibilities increased to include other communicable diseases.

The COPC approach outlined in the introduction has been implemented in the Eastern Cape via teams of CHWs referred to as Ward Based Primary Health Care Outreach Teams (WBPHCOTs). The teams can assist with community screening services for TB and also assist patients with adherence to therapy [27]. The coverage they provide in communities makes them a potentially effective tool in providing active surveillance for TB [31]. These teams are led by professional nurses with a mandate to provide comprehensive primary health care services to defined geographical areas [25,26]. Each team includes five to six CHWs and sometimes a health promoter. Their services are meant to be integrated with facility-based primary care [16].

## Selection of participants

The study intended to interview all 12 TB managers in the Province as described above and therefore there was no sampling and the interpretation of data was based on the entire study population of interest.

## Data collection

Invitations and arrangements for the interviews were made via email and telephone. The interviews occurred between May and September 2018. An interview guide was developed to explore current approaches to identifying cases of active TB, the effectiveness of these approaches and the ideal approach to identifying active TB. The researcher included additional questions or adapted existing questions based on responses from earlier interviews to ensure adequate exploration of the emerging themes and study objectives. Most semi-structured interviews were conducted in the offices of the TB managers and three interviews were conducted telephonically. All interviews were conducted in English, did not require translation and were digitally recorded.

## Data analysis

The recorded interviews were transcribed verbatim by a professional transcriber, before commencing a thematic analysis with the framework method using ATLAS ti software [36]. This process involved becoming familiar with the data by reading transcripts and listening to tapes of the recorded interviews. The researcher then developed, by an inductive approach, a thematic index of codes and categories. This was followed by coding all the transcribed data as per the thematic index, during this process some additional codes were developed. All the similarly coded data were collated into charts based on the categories of the thematic index. The researcher then interpreted the charts for the key themes and subthemes, and any relationship between themes. The creation of the thematic index and data interpretation was an iterative process between the two authors.

# Findings

Ten TB managers were interviewed, which included three provincial managers and seven district managers, inclusive of the two Metros. One provincial manager declined to be interviewed and one district manager was not contactable. One interview was conducted with two people in attendance at the request of the HAST manager who believed herself not sufficiently knowledgeable on TB matters. The managers were all experienced female nurses, some with training in management, ranging between 35 and 65 years of age. In preserving anonymity in this small group of study participants the researcher will not be providing more personal details.

Themes are presented under the following broad categories that align with the study objectives: current approaches to screening for patients with active TB disease, the effectiveness of these approaches to TB screening and the ideal mode of screening.

## Current approaches to screening for patients with active TB disease

Managers identified a number of current approaches to screening of patients for active TB in health care facilities and communities. In primary care facilities, approaches ranged from screening every patient for TB symptoms to a focus on specific target groups:

"*We are screening at the facilities, that is in the clinics and hospitals, those that are not on TB treatment, especially those that are high risk group like your HIV positive clients, like your patients on chemotherapy. Like your diabetic clients and those with, um, that are very ill, ja like your cancer clients. And also, the under-fives, we are screening them.*"

(*Interview 9, mainly rural district*)

Hospitals screened all ambulatory and admitted patients for symptoms in order to find patients missed in the primary care services.

Managers preferred an approach that included screening at all levels of the health system. In the rural areas there was more interest in innovative approaches to community-based screening as access to facilities was more difficult, for example through monthly mobile clinic services on farms for seasonal workers. In the urban areas, managers were inclined towards screening in facilities, although they had more resources in the community to assist with screening. Urban areas had more facilities, but also a higher workload. This workload was due to the population density as well as rural people who bypassed their local rural facilities.

In some districts, the managers expressed a reliance on non-governmental organisations for community-based screening services. These organisations often employed CHWs to provide very specific services. These managers believed in the value of early identification in high risk communities. Even when they lacked the resources of a well-equipped WBPHCOT, they used NGOs and other partners to provide screening services in the community:

'*We are assisted by getting to outreach services first. I've got partners that are assisting me conducting door to door services, screening to find the new clients..*'

(*Interview 8, semi-urban district*)

In one district a NGO was mandated to conduct house to house screening services for TB and HIV. The WBPHCOTs on the other hand provided a range of services, inclusive of screening services for communicable and non-communicable diseases, preventive services across all age groups, and in some instances management of minor complaints. In some districts, there was a duplication of functions between the WBPHCOTs and the NGOs, while a number of rural districts lacked any NGO services. Nevertheless, in districts with overlapping services there was no reported evidence of more effective identification of patients with active TB. A sizable number of the identified TB patients were still found in the health facilities. Many of the NGO-led services were highly focused, for example only tracing people who were non-adherent with TB treatment or only screening for those at high risk of TB. Screening in schools was the responsibility of primary care nurses employed by the district to run the school health program.

Other strategies, which were used by most districts, included TB case-finding campaigns, often at public gatherings throughout the year and funded by government. The campaigns were often organised by the WBPHCOTs in response to data collected in the facilities or by the outreach teams that identified hotspots. The teams chose a point in the community from which they provided services during these campaigns and could collect specimens for investigation as required. One of the managers also mentioned the use of private general practices to screen for TB:

"*In all the areas now and again there are activities like going to the communities, like schools, like where we see it as a hotspot, the health promotion and the community health workers, they go even in households and they visit homes and they screen clients there*"

(*Interview 2, provincial manager*)

## Effectiveness of current approaches to TB screening

The managers identified a discordance between the numbers of people screened positive for symptoms in the community and the numbers of patients investigated for TB at the facility. It was unclear how the records monitored successful referrals and ensured follow up of all those that were identified with symptoms of TB. At the provincial level, the general opinion was that more needed to be done within the hospitals to identify TB. One provincial manager queried the relevance of active surveillance for TB in communities, when not enough was being done within facilities. This opinion seemed to result from the impression that resources were inadequate for community level screening and the focus therefore should only be on contact tracing for patients identified with TB in the facility. However, a number of managers at the district level opined that the WBPHCOTs were accepted by communities and provided good coverage, despite the challenges they faced:

"*Then there are those people who don't go to facilities, the very fact that when we do the activities it happened that we get some people who are positive. It means that if the opportunity was more, we would get more. But the problem is we don't have community services resources*"

(*Interview 2, Provincial Manager*)

Managers had doubts about achieving their goals for TB screening and thought the main problem was that stakeholders needed to coordinate their activities better, rather than working in isolation or in parallel. Although one manager was convinced the WBPHCOTs alone would enable her to meet the goals, others developed additional strategies. These ranged from community education during 'imbizos'(meetings) set up by local political leaders to the use of additional community-based organisations to expand coverage of TB screening services:

"*We are not going to meet the global target of ending TB by 2030 if we do notchange our strategies..*"

(*Interview 7, urban District*)

"*We check now the, the, if* (TB) *incidents, is it going high or is it going low. For example, if I can mention we have seen that at least now our numbers now are decreasing because of this active case finding that we are doing.*"

(*Interview 9, rural District*).

The factors contributing to the spread of TB in communities were not all under the control of the health services. The districts that identified this challenge developed a concept of 'war rooms' where different government departmental representatives collaborated to develop ideas for combatting these factors in the communities. The feedback was largely positive because this was an integrated approach to address the social determinants of health and could complement more effective screening.

Managers were convinced that the myriad of challenges identified with the implementation of the WBPHCOTs had a direct impact on the effectiveness of active surveillance for TB. A number of challenges were common to both the urban and rural health districts. These challenges included a lack of community awareness of the purpose for WBPHCOTs and of the TB epidemic, a lack of team leaders, inadequate transport resources, as well as a lack of coordination between the services provided by the NGO-funded CHWs and the government-funded WBPHCOTs. In some areas, there was an overlap between the two with some households

having more than one organisation visit for services, while some areas had no services at all. The CHWs in some settings were not adequately trained and were allowed to provide the service before completion of their training due to the pressure for service delivery. However, a provincial level manager mentioned the development of a formal training and mentoring program through the Eastern Cape Department of Health (ECDoH) to address this challenge.

Managers mentioned issues with recall of patients who provided the wrong address, had no formal address, or fell outside the area served by the WBPHCOT. One manager resolved that this was where she could use local community-based organizations (CBO), as they were more familiar with the community (a number of CHWs in WBPHCOTs were not living in the communities where they served). The different managers had different approaches and none seemed able to resolve this challenge fully. In some areas tracing patients was a responsibility of the WBPHCOTs, in other facilities this was the responsibility of whatever NGO supported the facility. NGOs differed from the local CBOs in that the CBOs were local to their communities and very focused on addressing needs of the local community, while the NGOs were bigger organisations often with international funders.

The development of service level agreements with the NGOs by people far removed from the sites of implementation meant that these agreements did not always fully address the needs in the districts. As a result, the services of the NGOs were not always aligned with the need for active surveillance for active TB. A number of NGOs had a very focused objective and adhered to this strictly as per their service level agreement, even when the need at the facility was different.

In the rural districts, the lack of transport for WBPHCOTs was more of a burden due to the topography that made access to home steads difficult. In some, there were no road networks and the home steads were kilometres away from each other. For a WBPHCOT, getting to each homestead could be hours of walking, with more time spent walking than providing the service:

> "*Another thing, which is a challenge in our areas, our areas are deep rural. They are too vast. There are people who built their houses up there and you cannot go there by foot.*"

> (*Interview 3, provincial manager*)

Security was a challenge for the WBPHCOTs in urban districts. These areas were associated with alcohol and drug abuse, high rates of unemployment, poverty, inadequate housing and sanitation. In these areas, WBPHCOTs were soft targets for opportunistic crime, which disrupted or prevented the service:

> "*They are attached to certain wards, and some wards or some areas is very rife with gangsterism, so violence is one of the reasons why they sometimes can't go out and do this active surveillance that we want them to assist us with*"

> (*Interview 7, urban District*)

Managers mentioned the stigmatization of people with TB by communities, particularly because of the association with HIV, which resulted in people not wanting visits from CHWs. One manager had engaged the local political and community leaders to allow CHWs slots during the community 'imbizos' to speak with members of the community. This community engagement may also have increased understanding and acceptance of the CHWs.

### Ideal model of screening for active TB

All managers strongly believed that facility-based screening was a key component in achieving the goals of the NTP:

"*Also the issue of screening, of course. What we must do is screening in our facilities, in each and every client that visits the facility, of which we are still far to get the 90% of the screened*"

(*Interview 1, urban district*)

The District TB program managers thought, however, that community-based screening via the WBPHCOTs was also essential. They thought, therefore, that WBPHCOTs should continue to assist facilities with tracing contacts of known TB patients:

"*I believe doc, that since we have poor outcomes within the Metro, when we look at our TB data, I believe that active surveillance needs to be incorporated within the package that is given to WBPHCOT. I believe so, because like I was saying earlier on, we need to have another strategy.*"

(*Interview 7, urban district.*)

Managers thought about innovative strategies that could be included in a COPC approach. Community leaders should be engaged to support active TB surveillance and encourage participation by community members in initiatives of the WBPHCOTs. WBPHCOTs should not only screen in households, but also focus on other opportunities such as screening seasonal farm workers, men who congregate in taverns and taxi ranks, people that attend churches, and at other public events in the most affected communities.

A majority believed the ideal approach to identifying patients with active TB should be a combination of active surveillance in communities incorporated into a comprehensive service delivered by WBPHCOTs and active case finding within facilities. The managers who indicated their dissatisfaction with current approaches to community based screening services still mentioned these as ideal, giving the impression that their dissatisfaction was not so much with the method as with implementation.

When asked to comment on ideal approaches to monitoring of active surveillance for TB in communities, managers wanted to maintain the current monitoring tools provided by the Department of Health. These tools were designed to measure the numbers screened for TB, numbers investigated for TB, numbers tested positive for TB and numbers started on anti-TB treatment. This provided enough information, although all work done in the WBPHCOT is included in the facility's report and cannot be disaggregated to show the contribution of the WBPHCOT alone. However, specific targets should be set for repeat screenings in each household.

The key findings from the study are summarised in Table 1.

## Discussion

### The case for implementing active surveillance for TB

The managers believed that active surveillance for TB should be performed in high risk communities in conjunction with active case-finding in health care facilities. This opinion is similar to the recommendations from the WHO that support a need for active surveillance [37]. The evidence from various high burden settings, similar to the context described by these TB managers, suggests a pool of active TB cases that remain unidentified in communities [38,39].

**Table 1. Summary of the key findings.**

**Current approaches to TB surveillance**:

1. Screening patients at hospitals and primary care facilities

2. Screening in communities via CHWs employed by NGOs

3. Screening in communities via CHWs as members of WBPHCOTs

4. Screening by healthcare workers via campaigns at public events

5. Use of CHWs to trace patients that were non-adherent to treatment and close contacts identified by the facility

6. Screening in private GP practices

**Effectiveness of current approaches to TB screening**

| **Challenges**: | **Successes**: |
|---|---|
| 1. Lack of coordination between different organisations and approaches | 1. Adequate coverage by WBPHCOTs in some districts. |
| 2. Inequitable distribution of resources between rural/ urban areas | 2. Development of innovative means to improve coverage–community education through 'imbizos', targeted screening initiatives, the use of local political leaders to promote coverage in communities, using community based organisations for active surveillance in communities |
| 3. Duplication of services between NGOs and WBPHCOTs in some communities | |
| 4. Gap between CHWs identifying people at risk of TB and actually being investigated for TB | |
| 5. Lack of community education about the TB epidemic | 3. Collaboration between various government departments in 'war rooms' to address the underlying factors contributing to the spread of TB in communities |
| 6. Stigmatisation of TB in communities | |
| 7. Development of health programs and service level agreements by people far removed from the site of problems | |
| 8. Difficulty retaining outreach team leaders in rural districts | 4. Development of a uniform training curriculum and mentoring opportunities for all CHWs through the ECDoH. |
| 9. Difficulty accessing certain rural communities due to topography | |
| 10. Security challenges in large towns and Metros in communities with alcohol and drug misuse and high unemployment rates | |

**Ideas on the ideal approach to active surveillance of TB**:

1. Integrate and coordinate a multifaceted approach that includes both facility-based and community-based screening

2. Rational geographic delineation of community-based services between NGOs and WBPHCOTs to maximise use of resources.

3. Work with leaders in the community to strengthen community engagement and participation

4. Setting specific targets on frequency of household screening in the community and the use of monitoring tools based on recommendations of the National Department of Health.

According to the managers, TB was more prevalent in disadvantaged communities. Social determinants of health have a direct impact on the epidemic in any community and so there is a need for a variety of community level interventions that respond to specific determinants feeding the epidemic, similar to the 'war room' concept mentioned by the managers [20,40,41].

The current approaches to identifying active TB patients in this context appear to rely heavily on facility-based active case finding strategies. However, patients that present to facilities may have more overt signs and symptoms or better access and can be presumed to represent only a fraction of all patients with active TB [38]. The overwhelming impression was that managers were constrained by limited resources and needed to choose the more cost-effective options. However, research in similar resource constrained settings show the medium to long term effects of community-based screening strategies can contribute more to reducing the incidence of TB and mortality associated with the epidemic [23,31]. However, some research

reports that although active surveillance increases the yield of new TB cases this does not necessarily translate into improved treatment outcomes [17,42,43]. This may explain why some managers think it is better to concentrate their resources on achieving the best outcomes for patients already identified. Their dilemma was whether to use their limited resources for community-based surveillance via WBPHCOTs or focus on facility-based case-finding strategies.

The cost-effectiveness of community-based surveillance would be increased by bridging the gap between patients identified in the community and the numbers actually investigated for TB in facilities [31,44]. One possible strategy is for CHWs to collect sputum in the home rather than relying on people to access TB investigation at the local facility [45,46]. Current findings indicate this service is provided by some CHWs working for NGOs. The services provided by WBPHCOTs may also be more cost-effective because they are more comprehensive and integrated services, with active TB surveillance only one of the activities during a household visit [47–49]. As opposed to contracting with CHWs and NGOs to deliver TB specific services.

## Successes and challenges with implementing active surveillance for TB

The findings in this study suggest that a lack of knowledge about TB in communities and among CHWs remains a barrier to implementing active surveillance. Findings in the literature suggest that improved awareness in the community and amongst CHWs results in a decrease in diagnostic delay [44]. This barrier is already been addressed through the training opportunities afforded CHWs by the ECDoH. In addition to this, the recent NDoH policy document recommends that all new CHWs should have successfully completed high school, suggesting a move towards improved capability of CHWs providing COPC [16]. Some of the managers are increasing awareness about TB in their communities through community engagement. Community engagement is recognised as a strategy for health promotion as it empowers communities to take responsibility for their own health [47]. The extent of community engagement was not a focus of this study.

The implementation of COPC in communities reveals a number of challenges [25]. These challenges are often associated with allocation of resources, adequacy of supportive supervision, sufficient training, the acceptability of teams to local communities and in some instances the understanding of CHWs' roles [25,26,48]. In the urban centres, security challenges made it difficult to provide community based services in crime-ridden communities. These challenges are not unique to the Eastern Cape Province [48]. Research in other centres with better established WBPHCOTs recommend recruitment of local community members on the teams. In the worst affected areas, the WBPHCOTs simply avoided delivering services. The impact is that these areas become high burden TB settings with poor health outcomes. In the rural areas, transportation and staffing challenges were more of a challenge. The staffing challenges were often a result of factors outside of the health system, no adequate schools for children of staff or no jobs available for the spouse. Again, these are not unique to the Eastern Cape, [25,26] however, the fact that the Province is rural and covers a large land area is a further limitation to how efficiently the CHWs run their service without team leaders in a number of circumstances. In response to staffing challenges the NDoH policy document approved the use of enrolled nurses, oriented to community health, as team leaders.

In other SA settings challenges have not resulted in turning away from active surveillance, but rather finding alternative ways to deliver the screening services [25,26]. The current policy according to the managers is to avoid hotspots where crime is rife. The current policy framework makes no mention of the process for response to the security challenge, however community participation and collaboration could assist with addressing this challenge [16].

Independent NGOs with international funds, despite service level agreements with the DoH, have a set agenda and goals of the funder tend to define service delivery initiatives [44]. This may explain the very narrow focus described by the managers in their experience with the NGOs. The WBPHCOTs have the potential to provide a more comprehensive and directly accountable service with good coverage based on a standard package of services [16]. On the other hand, NGOs can be more innovative (e.g. allowing collection of TB specimens in the community), with better community engagement, while aligned with a narrower focus on specific services such as tracing close contacts or patients that are not adherent to their treatment. NGOs may also be well established, tend to only employ CHWs derived from the local community and have more supportive supervision. The WBPHCOTs are often newly established, may employ CHWs from outside the community and struggle to recruit and retain team leaders [49]. It certainly makes little sense to duplicate services in the same community. The challenges of resource constraints and the need for efficient use of resources means utilising WBPHCOTs and NGO services in a well-coordinated and geographically contiguous manner [25,27,45,48,50].

Community engagement, aimed at empowering communities to participate in activities that promote wellness, is one of the strategies proposed in the Ottawa charter [51]. Community engagement can help drive appropriate change in health systems, enhance community awareness of health problems and enable collaboration on programmes to improve health [34]. CHWs can contribute to this and help bridge the gap between communities and their primary care facilities [35]. According to the WHO, community engagement also increases awareness and acceptance of CHWs and improves their safety [52]. Managers in this study alluded to the need for more engagement with community leaders and campaigns to reduce TB transmission. CHWs can help enable such collaboration with community members and political leaders and improve health outcomes in the community [53]. A few of the managers described such interventions in their district, although this was not widespread or well developed.

Managers were also aware of the importance of addressing the social determinants of health through intersectoral collaboration. This could reduce the incidence of TB and the need for such active surveillance. This can only function in a setting where government departments collaborate [54,55]. While the DoH should actively pursue such intersectoral collaboration the implementation of active surveillance for TB is more in their direct and immediate circle of control. Such an intersectoral approach to community health needs is also consistent with the philosophy of COPC [16,25,56]. The WBPHCOT model may work best if it fully embraces a COPC approach and does not regard itself as just an outreach from primary care facilities [27,57].

## Limitations of the study

This was a qualitative study in the context of the Eastern Cape Province and although highly contextual, some findings may be transferrable to similar settings in South Africa. Two managers were not interviewed, their views could have added additional themes or more depth of understanding. Only one researcher performed all the steps in data analysis, albeit in an iterative process with the second researcher with regard to the thematic index and final interpretation. The inclusion of a second independent researcher in data analysis might have strengthened the analysis. Triangulation of the data was not possible and respondent validation was not performed.

Further descriptive qualitative exploration of active TB surveillance projects elsewhere in the country is planned. The findings from this paper and these other projects will be integrated

and the issues identified will be incorporated into a descriptive survey of healthcare workers in the Eastern Cape province. This will help to quantify the findings in clinical practice and lessons learnt will inform a quality improvement project in one of the sub-districts.

## Recommendations

Based on the findings of this study we recommend that active surveillance for TB should:

- Include a multifaceted facility- and community-based approach, with more resources targeted at strengthening WBPHCOTs, for whom this would be part of a comprehensive and integrated service in a COPC framework.

- Be strengthened through continued standardised training of WBPHCOTs and supportive supervision with provision of transport where required.

- Be supported by adequate community engagement and collaboration to improve the security of CHWs through increased acceptance and understanding of their roles by the community.

- Evaluate the effectiveness of CHWs collecting samples in the home in order to reduce the gap between screening and facility-based investigation.

- Be enhanced by rationalising the footprint of similar services offered via WBPHCOTs and NGOs to avoid duplication in the same communities.

## Conclusion

Active surveillance for TB in the community was an ideal that most managers supported. Effective surveillance was limited by budgetary constraints, but also by inefficient coordination of existing resources that were divided between government and non-government organisations. Gaps existed between identification in the community and diagnosis at facilities. WBPHCOTs were limited by systemic factors that also differed between rural and urban areas (e.g. transport, security, standardised training). Implementation of active surveillance needed to be complemented by better community engagement and collaboration as well as attention to the underlying social determinants via inter-sectoral 'war-rooms'.

## Supporting information

**S1 File. Interview guide for TB managers.**
(PDF)

## Acknowledgments

Special thanks to Dr E.E. Ajudua and managers of the TB program in the Eastern Cape of South Africa.

## Author Contributions

**Conceptualization:** Febisola I. Ajudua.

**Data curation:** Febisola I. Ajudua.

**Formal analysis:** Febisola I. Ajudua, Robert J. Mash.

**Funding acquisition:** Robert J. Mash.

**Methodology:** Febisola I. Ajudua.

**Project administration:** Febisola I. Ajudua.

**Supervision:** Robert J. Mash.

**Writing – original draft:** Febisola I. Ajudua.

**Writing – review & editing:** Febisola I. Ajudua, Robert J. Mash.

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
