## [Decision Letter · Decision Letter 0]

5 May 2020

PONE-D-20-02975

Implementing active surveillance for TB – The views of managers in a resource limited setting, South Africa.

PLOS ONE

Dear Dr Ajudua,

Thank you for submitting your manuscript to PLOS ONE. After careful consideration, we feel that it has merit but does not fully meet PLOS ONE’s publication criteria as it currently stands. Therefore, we invite you to submit a revised version of the manuscript that addresses the points raised during the review process.

We would appreciate receiving your revised manuscript by Jun 19 2020 11:59PM. To enhance the reproducibility of your results, we recommend that if applicable you deposit your laboratory protocols in protocols.io, where a protocol can be assigned its own identifier (DOI) such that it can be cited independently in the future. For instructions see: http://journals.plos.org/plosone/s/submission-guidelines#loc-laboratory-protocols

We look forward to receiving your revised manuscript.

Kind regards,

Petros Isaakidis

Academic Editor

PLOS ONE

Journal Requirements:

2. Please include additional information regarding the interview guide used in the study and ensure that you have provided sufficient details that others could replicate the analyses. For instance, if you developed a guide as part of this study and it is not under a copyright more restrictive than CC-BY, please include a copy, in both the original language and English, as Supporting Information.

Reviewers' comments:

Reviewer's Responses to Questions

**Comments to the Author**

1. Is the manuscript technically sound, and do the data support the conclusions?

Reviewer #1: Yes

Reviewer #2: Yes

2. Has the statistical analysis been performed appropriately and rigorously? 

Reviewer #1: Yes

Reviewer #2: N/A

3. Have the authors made all data underlying the findings in their manuscript fully available?

Reviewer #1: No

Reviewer #2: Yes

4. Is the manuscript presented in an intelligible fashion and written in standard English?

Reviewer #1: Yes

Reviewer #2: Yes

5. Review Comments to the Author

Reviewer #1: Overall this is a well written and interesting qualitative study which aimed to assess manager’s views on the implementation of active surveillance for TB. This is addressing a gap in the literature as there have not been many studies to date which have investigated this. Additionally, the results can potentially drive changes in clinical practice and ensure that resources are streamlined in order to ensure effective service delivery.

General Comments:

Kindly avoid the use of stigmatizing language i.e. defaulter, suspected TB case.

Throughout, ensure that you double check your grammar. There are multiple instance of incorrect grammar.

Introduction:

In the introduction section you should mention that TB is THE LEADING cause of mortality in South Africa

This section, while very interesting is quite long. I recommend that you try to make this more concise. As noted below, some of the information provided in the introduction describing the programme might be better suited in the methods section.

Methods:

In the section on the setting you might want to consider providing some information on the burden of TB and the coinciding HIV prevalence. Additionally, if you have any information on the average income/housing situation there this might help to better contextualize the context.

In the section on the data collection, you might want to specify what language the interviews were done in and if they required any translation. Additionally, it might be useful to provide some information on the questions asked. You could consider including the questionnaire in an annex. Overall, without being able to review the semi-structured questionnaire it is hard to understand the questions asked and what further areas were investigated. I think that sharing this would help future research carry out similar investigations or make the needed adaptations to them.

Lines 152-160 could be included under a subheading titled programme description. Then I think some of the details regarding the programme provided in the introduction section could be moved here.

Discussion:

I recommend that you include the summary of the key findings in the results section rather than the discussion section. Maybe you could do that in the beginning of the results section and then unpack each point further in the subsections as you have already done. This could for example be placed after line 203.

In the discussion check your sentence on lines 451-52; it does not make sense.

On line 463 is there a more recent document you can reference.

Paragraphs on 433-441 and 469-482 both talk about security risks and challenges as a result. You might want to consider combining these paragraphs to avoid redundancy.

On lines 492-499 it seems as though you are discussing some of your imitations. However, I recommend that you specifically stipulate that the study is subject to limitations which include XXXXXXX. You have included all the relevant information here, I just think that it needs to be slightly reframed.

Reviewer #2: The qualitative study entitled “Implementing active surveillance for TB – The views of managers in a resource limited setting, South Africa” addresses the critical question of what frontline TB program managers see to be key challenges and solutions in implementing active surveillance for TB. The study interviewed 10 of 12 Eastern Cape provincial TB managers. The authors conclude that a multi-faceted facility and community-based approach would be most effective for active case finding.

This paper addresses an important question that informs TB policy and the research design and qualitative methods are appropriate for the research question. However, I recommend that the presentation of the interview data be more structured and streamlined to increase the contribution of the paper to our understanding of TB program design.

The paper claims to present “innovative” ideas for improving active surveillance, however it’s not clear that the qualitative interviews provide a lot of new information on these topics that hasn’t already been covered in the literature. The introduction should perform a more thorough literature review to identify gaps in our understanding of how to design active surveillance for TB.

Table 1 clearly summarizes the results in an accessible way. I suggest using a framework like this for the rest of the paper. In its current form, the results and discussion sections meander a little through the responses to questions rather than concisely laying out the main themes from the interviews and illustrative quotes to accompany them.

Did the interviews ask about prioritization of the different innovative ideas? The solutions are presented as a long list, which makes it hard for resource-limited programs to identify the highest-impact components to implement. Is there a way to infer some sort of prioritization from the frequency or intensity of themes raised by interviewees?

The conclusion that “this study suggests a multi-faceted facility and community-based approach would be most effective for active case finding” is quite general, especially considering it’s based on direct interviews with TB program managers. I suggest combing through the results in this paper to identify a more significant contribution to our knowledge that this study brings to light.

Both the introduction and the discussion section are quite long. I’d suggest streamlining them to convey the main points more efficiently which will increase readership of the article.

The quotes included in the article do not add an awful lot of color or information to the paper. I’d suggest including a table with several shorter quotes on each topic that focus on key phrases related to the study themes. Perhaps the discussion could weave more detail around the quotes to make it clear what each quote brings to light.

6. PLOS authors have the option to publish the peer review history of their article (what does this mean?). If published, this will include your full peer review and any attached files.

Reviewer #1: No

Reviewer #2: No

---

## [Author Response · Author response to Decision Letter 0]

24 Jul 2020

Thank you for the critique of this paper. The revisions advised have been considered and appropriate changes made. Please find attached in supporting documents the comprehensive response to reviewers.

---

## [Editor Report · Decision Letter 1]

4 Aug 2020

PONE-D-20-02975R1

Implementing active surveillance for TB – The views of managers in a resource limited setting, South Africa.

PLOS ONE

Dear Dr. Ajudua,

Thank you for submitting your manuscript to PLOS ONE. After careful consideration, we feel that it has merit but does not fully meet PLOS ONE’s publication criteria as it currently stands. Therefore, we invite you to submit a revised version of the manuscript that addresses the points raised during the review process.

While I see that the authors have adequately responded to most of the reviewers' comments and suggestions and have satisfactorily revised their manuscript, I particularly noticed that they have not addressed the issue of stigmatizing language and terms, such a defaulters. I also recommend that authors use the COREQ checklist, or other relevant checklists listed by the Equator Network, such as the SRQR, to ensure complete reporting.

We look forward to receiving your revised manuscript.

Kind regards,

Petros Isaakidis

Academic Editor

PLOS ONE

---

## [Author Response · Author response to Decision Letter 1]

3 Sep 2020

Thank you for the extensive review of this paper. We have included as requested the COREQ checklist for reporting qualitative research. We have removed all stigmatizing language in the paper. Thank you.

---

## [Editor Report · Decision Letter 2]

7 Sep 2020

Implementing active surveillance for TB – The views of managers in a resource limited setting, South Africa.

PONE-D-20-02975R2

Dear Dr. Ajudua,

We’re pleased to inform you that your manuscript has been judged scientifically suitable for publication and will be formally accepted for publication once it meets all outstanding technical requirements.

Kind regards,

Petros Isaakidis

Academic Editor

PLOS ONE
---

## [Editor Report · Acceptance letter]

17 Sep 2020

PONE-D-20-02975R2

Implementing active surveillance for TB – The views of managers in a resource limited setting, South Africa.

Dear Dr. Ajudua:

I'm pleased to inform you that your manuscript has been deemed suitable for publication in PLOS ONE. Congratulations! Your manuscript is now with our production department.

Kind regards,

on behalf of

Dr. Petros Isaakidis 

Academic Editor

PLOS ONE